# An Ultrasonic RF Acquisition System for Plant Stems Based on Labview Double Layer Multiple Triggering

**DOI:** 10.3390/s23167088

**Published:** 2023-08-10

**Authors:** Xin Huang, Danju Lv, Rui Xi, Mingyuan Gao, Ziqian Wang, Lianglian Gu, Wei Li, Yan Zhang

**Affiliations:** 1College of Big Data and Intelligent Engineering, Southwest Forestry University, Kunming 650224, China; huangxin615@swfu.edu.cn (X.H.); xirui@swfu.edu.cn (R.X.); tuayuan@swfu.edu.cn (M.G.); wangziqian@swfu.edu.cn (Z.W.); gulianglian@swfu.edu.cn (L.G.); liwei8152@swfu.edu.cn (W.L.); 2School of Mathematics and Physics, Southwest Forestry University, Kunming 650224, China; zhangyan@swfu.edu.cn

**Keywords:** ultrasonic RF, double layer multiple triggering, data acquisition, LabVIEW, plant stem body

## Abstract

Ultrasound is widely used in medical and engineering inspections due to its non-destructive and easy-to-use characteristics. However, the complex internal structure of plant stems presents challenges for ultrasound testing. The density and thickness differences in various types of stems can cause different attenuation of ultrasonic signal propagation and the formation of different echo locations. To detect structural changes in plant stems, it is crucial to acquire complete ultrasonic echo RF signals. However, there is currently no dedicated ultrasonic RF detection equipment for plant stems, and some ultrasonic acquisition equipment has limited memory capacity that cannot store a complete echo signal. To address this problem, this paper proposes a double-layer multiple-timing trigger method, which can store multiple trigger sampling memories to meet the sampling needs of different plant stems with different ultrasonic echo locations. The method was tested in experiments and found to be effective in acquiring complete ultrasonic RF echo signals for plant stems. This approach has practical significance for the ultrasonic detection of plant stems.

## 1. Introduction

Ultrasonic inspection [1,2] is a non-destructive testing method that utilizes ultrasonic waves to detect defects within objects and on their surfaces by leveraging their acoustic properties. This technique is extensively employed for evaluating the dimensions, geometry, and distribution of defects, and is one of the most widely adopted non-destructive testing methods. In comparison to other non-destructive testing techniques [2], ultrasonic inspection exhibits remarkable versatility, user-friendliness, human safety, cost-effectiveness, precise defect localization, high sensitivity, and is suitable for field operation [3].

Data acquisition systems play a crucial role in measurement and control systems and are widely employed in modern industry [4]. Virtual instrumentation combines computers and instruments, offering flexibility and versatility [5,6,7], thereby meeting the diverse data acquisition needs of various inspection systems [8]. For instance, Haijun [6] developed a PC-based virtual instrumentation system using the LabVIEW graphical programming language and a high-digitization-rate A/D sampling card (PCI-12400) to calculate ultrasonic velocity and attenuation coefficient from the acquired ultrasonic echo signals; Changyun [5] designed an ultrasonic detection system utilizing TMS320C5402 as the hardware for acquiring, analyzing, processing, and storing ultrasonic echo signals; Roy [7] utilized the LabVIEW platform to measure the propagation time of ultrasonic waves between a transmitter and receiver, enabling non-contact measurement of liquid density; Ning [9] constructed a real-time online test platform for ultrasonic motor impedance characteristic testing by integrating LabVIEW and FPGA. Puantha [10] employed Arduino and LabVIEW to measure the sound velocity of pipe resonance in the air; Stefenon [11] evaluated the automation capability of conventional grid insulator analysis using LabVIEW (2014) software by connecting the ultrasound detector’s generated audible noise to a personal computer and obtaining the FFT signal. Guo [12] used sensors to acquire human heart sounds and ECG signals on the LabVIEW virtual instrument platform, achieving the segmentation and localization of heart sounds and ECG signals.

Existing memory settings of dedicated ultrasound acquisition devices are limited by the echo position range of the detected object and rely on a fixed number of sampling points determined by the acquisition card’s capacity. However, in the field of ultrasound analysis of plant stems, where the detection of ultrasound radio frequency (RF) signals is a novel research area, there are unique challenges and potential applications for non-destructive, real-time monitoring of cavitation, embolization, and stem moisture changes through RF signal analysis [13,14].

In practical ultrasonic inspections of plant stems, two primary challenges arise. Firstly, there are variations in stem thickness, and secondly, the structural differences in the stem affect the propagation speed of ultrasound. These factors result in a wide range of variations in the echolocation of plant stems, making it challenging to determine a fixed memory capacity for storing the number of points sampled in a single echo.

To address these challenges, this paper proposes a solution for ultrasound inspection of plant stems. The solution involves the development of a plant stem ultrasound acquisition system that employs a double-layer multiple trigger sampling approach. This innovative system allows for the dynamic adjustment of the number of ultrasound pulse acquisition points, effectively accommodating the diverse sampling requirements arising from varying thicknesses and structures of plant stems.

The core component of the acquisition system is Fanhua’s Nextkit high-speed acquisition card, while LabVIEW is utilized as the upper computer for real-time acquisition, analysis, processing, and storage of ultrasonic signals. The portability and flexibility of the system enable comprehensive acquisition, thereby resolving the issue of incomplete storage of echo RF signals due to limitations in memory capacity caused by edge-triggered sampling.

In summary, this proposed solution aims to overcome the challenges faced in ultrasound inspection of plant stems by introducing a novel ultrasound acquisition system capable of adapting to the varying echolocation characteristics of plant stems. The system’s ability to dynamically adjust the number of acquisition points contributes to more accurate and comprehensive data collection, enabling improved analysis and understanding of plant stem characteristics.

## 2. System Composition

The ultrasound frequency-emission acquisition system comprises two primary components: hardware and software. As illustrated in Figure 1, the hardware system includes an ultrasonic RF generator, a receiver, a data acquisition card, and a PC. The ultrasonic RF generator and receiver emit ultrasonic waves through the probe and capture the corresponding echoes, which are subsequently transmitted to the computer via a high-speed acquisition card. On the other hand, the software system consists of a LabVIEW program running on the PC, offering functionalities such as data processing and storage.

## 3. LabVIEW-Based Double Layer Multiple Trigger Acquisition

### 3.1. Single-Layer Edge-Triggered Sampling Based on LabVIEW

Figure 2 illustrates the single-layer edge-triggered sampling process implemented in LabVIEW. The process consists of the following steps: (1) Initializing the data acquisition card by specifying the sampling channel ($N_{channel}$), sampling depth ($N_{card}$), and sampling rate ($N_{rate}$); (2) Configuring the virtual oscilloscope timing within LabVIEW and setting the sampling trigger mode. The rising or falling edge of the ultrasonic pulse signal serves as the trigger for data acquisition, which continues until the memory of the card is full, storing the acquired data on the PC. In this acquisition system, the ultrasonic pulse emission frequency (f) is set to 500 Hz, equivalent to a period (T) of 20 ms. The time required for the acquisition card to acquire and store 500 sampling points ($N_{samples}$) on the PC is significantly shorter than 20 ms. Consequently, the system effectively ensures that each triggered ultrasonic pulse’s sampling can be saved on the PC.

(1)Flowchart

**Figure 2 sensors-23-07088-f002:**
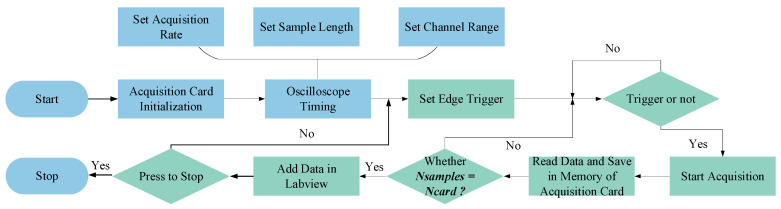
Single-layer edge-triggered sampling.

(2)Pseudocode (Algorithm 1)

**Algorithm 1:** Data Acquisition1Initialization $N_{rate}$ = 10 M/s, $N_{card}$ = 500 samples, $N_{channel}$ = CH1, Data, $N_{labview}$
2Set virtual oscilloscope timing
While button_stop == 1:3        IF length (Data) < $N_{labview}$:4                Trigger parameter setting: Edge_Trigger
5                IF Trigger: 6                        Acquisition data is stored in the memory of the acquisition card7                Else: 8                        Return Step 49        Data --> $N_{labview}$
10Data-->excel sheet

(3)Storage Problem

During the acquisition process, a challenge arises due to the storage length ($N_{card}$) of a single continuous acquisition trigger card being smaller than the required number of sampling points ($N_{pulse}$) for a complete ultrasonic pulse, resulting in incomplete acquisition. Specifically, when the frequency (f) of the ultrasonic pulse signal transmitted by the transmitter–receiver is set to 500 Hz, the corresponding pulse period (T) is calculated as 2 ms using Formula (1). In this scenario, assuming a sampling rate ($N_{rate}$) of 10 M/s, the total number of sampling points ($N_{pulse}$) required to capture complete information for a single ultrasonic pulse amounts to 20,000, as expressed by Formula (2).
(1)$$T=\frac{1}{f}$$
(2)$$S=T\times V_{samples}$$

Pulse edge-triggered acquisition is commonly employed to achieve accurate time synchronization during continuous pulse acquisition. However, the limited memory capacity ($N_{card}$) of the acquisition card poses a challenge when it is smaller than the required number of sampling points ($N_{samples}$) to capture a complete ultrasonic pulse. 

In this study, with $N_{card}$ set to 500 samples and an ultrasonic pulse frequency (f) of 500 Hz, the calculation based on Equation (2) yields $N_{pulse}$ as 20,000. Consequently, each pulse-triggered acquisition is capable of capturing only the initial 500 samples of the ultrasonic echo, leaving the remaining 19,500 signals unattainable.

### 3.2. Data Acquisition Based on LabVIEW Double Layer Multiple Times of Timing Triggering

(1)Flow chart

To address the issue of incomplete pulse sampling due to the acquisition card’s storage capacity ($N_{card}$) being smaller than the required number of sampling points ($N_{pulse}$), this paper proposes a double-layer multiple trigger approach for achieving comprehensive data acquisition of ultrasonic pulse RF signals. The acquisition process is illustrated in Figure 3.

(2)The principle of double layer multiple timing trigger

The double-layer trigger method presented in this study involves employing two distinct triggering techniques within a single pulse–echo acquisition. In the first layer of triggering, the rising edge of the input signal’s ultrasonic pulse is utilized as the trigger to ensure precise timing alignment of the acquired ultrasonic waves. This enables the acquisition and alignment of the initial 500 points of the pulse–echo, which are stored in the LabVIEW Data array. For the second layer of triggering, a soft triggering method utilizing dynamic timer delay is employed to achieve consistent data acquisition with a sampling depth of $N_{card}$ across various delays. This enables the complete sampling of a single ultrasonic pulse–echo. The sampling triggering process is illustrated in Figure 4 and Figure 5.

(3)Pseudocode (Algorithm 2)

**Algorithm 2:** Data Acquisition1Initialization $N_{rate}$ = 10 M/s, $N_{card}$ = 500 samples, $N_{channel}$ = CH1, Data, $N_{labview}$
2Set virtual oscilloscope timing3While button_stop == 1:4        IF length (Data) < $N_{card}$:5                Trigger parameter setting: Edge_Trigger
6                IF Trigger:7                     Acquisition data is stored in the memory of the acquisition card8        Else:9                  Return Step 510        Data --> $N_{labview}$
11        Trigger parameter setting: Soft_Trigger

        IF length(Data) < $N_{labview}$:12                STCount = 113                Timing soft trigger subroutine, get Timing_i
14                Tset_i = Timing_i ×STCount;15                IF Timing_i = Tset_i:16$N_{card}$ continuous data acquisition by capture card17                
STCount=STCount+1
18Return Step 1419$N_{card}$-->Data20Data-->excel sheet

(4)Implementation

The updated version of the acquisition program primarily consists of the first two layers of triggered acquisition. The implementation of the first layer involves the external pulse signal rising edge triggering, as demonstrated in Figure 6. The program calls the “nextkit_AI Timing.vi” and configures the trigger channel as channel 1. Within this VI, the “iTrigType” parameter is set to 0, indicating a rising edge trigger. Additionally, the program utilizes the “ReadData.vi,” which incorporates the functions TriggerEnable, StartAcq, GetFinishFlag, and ReadData. The “Channel 1 Data” output of the ReadData.vi is connected to the “Add Array” function (utilizing a shift register), with the corresponding add array control depicted in Figure 7.

The implementation of the second layer of multiple-timing triggers is presented in Figure 7. Initially, the trigger source is set to soft trigger by invoking the “nextkit_SetTrigger” library function from the DLL file. Subsequently, the “nextkit_StartAcq” function is called to initiate data acquisition. The soft trigger function, “nextkit_SoftTrigger,” is connected to the “GetFinishFlag” function, introducing a delay for the soft trigger based on the program’s runtime, as calculated in Figure 7. Once the acquisition is completed and the “nextkit_ReadData” function is called, the data is stored in a newly created “array subset”. The data from the array subset is then connected to the “add array” control. An “array size” control is added to determine the size of the current “add array”. If the current array is smaller than the specified number of data points, the current loop continues by using a shift register. If the condition is met, the program proceeds to the first-level acquisition section through the shift register, initiating the next round of data acquisition. This process repeats until the stop button is pressed.

The critical aspect of implementing sampling triggered by multiple dynamic time delays in the second layer lies in setting the delay time. Figure 8 demonstrates the procedure, which involves adding a time counter, Tinit, outside the while loop. Within the loop, a “subtraction” control is included, allowing the calculation of the difference between Tinit and a new time counter, Tend. By displaying this difference, the cumulative delay, Timing_i, can be obtained (Algorithm 3).


**Algorithm 3:** Timing soft trigger subroutineProblem solved: achieve dynamic delay of the acquisition window**Input**: Tinit**Output**: Timing_i
1

$Timing_{i}=Tend-Tint$

2Return Timing_i



## 4. Experiments

### 4.1. Double-Layer Multiple Trigger Sampling Verification

To validate the effectiveness of the proposed double-layer multiple trigger sampling, the CTS-8077PR ultrasonic RF generation receiver, equipped with a non-metallic probe, was employed to generate ultrasonic RF signals. These signals were then captured by the nextkit high-speed acquisition card developed using LabVIEW, as illustrated in Figure 9. Additionally, a DS1052E oscilloscope was utilized for comparison purposes.

Since the ultrasonic RF echo signal in the plant stem body, being a non-uniform medium, experiences significant attenuation, the system’s acquisition timing accuracy alone cannot ensure complete data acquisition. Therefore, this study conducted validation experiments using a homogeneous medium made of organic glass.

(1)Test sample: a plexiglass cylinder with a diameter and height of 6 cm. Its density was 1.18 g/${cm}^{3}$.(2)Parameter setting

Ultrasonic RF generation receiver parameters: pulse–echo frequency f = 500 Hz, receiving gain: +45 dB, ultrasonic probe frequency = 5.00 MHz. Acquisition device parameter setting: acquisition channel $N_{channel}$ is channel 1 (CH1). Sample length $N_{samples}$ = 500. According to the Nyquist sampling theorem, the sampling frequency $N_{rate}$ = 10 M/s and the edge trigger is set to rising edge. The Labview ultrasonic echo signal sampling array length $N_{pluse}$ is set to 1000 and 20,000 points, respectively.

(3)Experimental results

The experimental results are shown in Figure 10, Figure 11, Figure 12 and Figure 13. Figure 10 shows the experimental results of acquiring 500 $N_{samples}$ by single-layer rising edge triggering. Figure 11, Figure 12 and Figure 13 shows the expansion of 500 $N_{samples}$ into 1000 and 20,000 $N_{pluse}$ pulse–echo signals using double-layer multiple triggering. Figure 11 indicates that the ultrasonic first echo signal cannot be shown, i.e., the first echo position appears after 500 sampling points; Figure 11 indicates that the first echo position appears at the 2nd sampling; Figure 13 indicates that among the 20,000 $N_{pluse}$ echoes, it can be shown that the first echo position is at the sampling point and the second echo position is at the 3rd sampling, and the echo signal details are more complete. The above experiments show that: the double-layer multiple acquisition method proposed in this paper solves the problem of incomplete echo signal acquisition due to insufficient memory of the acquisition card.

To verify the accuracy of the experimental results, the echo signals acquired through the oscilloscope were compared, as shown in Figure 14, and the time to acquire the first echo signal using a double layer with multiple triggers was about 66.2 μs. As shown in Figure 14b, the DS1052E oscilloscope shows that the first echo time is 64.4 μs. The difference is 1.8 μs, and the difference in detection distance is 0.4914 cm, which is within an acceptable range.

### 4.2. Plexiglass Ultrasonic Acquisition Verification

The accuracy of timing plays a crucial role in the double-layer multi-trigger acquisition method as it relies on different timing trigger samplings. In this study, the accuracy of the echo signals collected by the ultrasonic acquisition system was verified by calculating the ultrasonic detection thickness based on the position of the first echo detected by the system. The calculation utilized the known propagation speed of ultrasound in plexiglass and was compared with the actual measured thickness.

Experimental setup included two plexiglass cylinders of equal diameter and height, measuring 6 cm and 10 cm, respectively. The detection was conducted in both axial and radial directions. The parameters for the detection process were as follows: the length of the ultrasonic echo signal sampling array was set to 2000 $N_{samples}$, the ultrasonic pulse–echo was collected ten times, and the remaining parameters were consistent with the aforementioned settings.

In accordance with the principle of ultrasonic detection, the depth *D* of the detected object can be determined using Equation (3):(3)D=vt
where *v* is the propagation velocity of ultrasound in the detected object. The test substance in this experiment is plexiglass, whose velocity is $v_{Plexiglas}=2730\;m/s$.

*t* is the time when the ultrasonic wave is received in the detected object as an echo, which is the time corresponding to the position of the first echo.

Figure 14 displays the time domain plot of the echo position in the homogeneous plexiglass. The acquired image is shown on the left, while the oscilloscope image is presented on the right. In the collected image, the X-axis represents time (in seconds), the Y-axis represents amplitude, and the position of the first echo signal is indicated. Conversely, in the oscilloscope image, the first echo position is denoted by T. It is important to note that due to the limited size of the oscilloscope screen, the display may not capture the second echo signal.

As shown in Table 1, the error range between the measured ultrasonic echo distance and the actual distance is 0.0125–0.0199 cm; the error range between the detected thickness of the axial and radial directions of the plexiglass and the actual thickness is 0.03–0.14 cm. By comparing the first echo time of the DAQ and oscilloscope, it can be seen that the error is between 0.01 microseconds and 0.07 microseconds. The error is acceptable for plant stem detection.

For further comparison, we exported the complete waveform collected by the oscilloscope, and performed Fourier transform on the two waveforms to obtain the spectrogram and phase diagram, as shown in Figure 15. In the axial direction, it can be seen that the frequency of the first echo is between 50,000 and 150,000 HZ, and the maximum amplitude is about 200, the frequency of the second echo is around 250,000 to 300,000 HZ, and the energy of the third echo is between 420,000 and 500,000 HZ. In the radial measurement, because the amplitude of the echo is too small, only faint energy can be seen at 70,000 to 150,000 Hz.

Figure 16 shows a plot of the number of collected data points versus the saving speed. The x-axis represents the total number of data points collected, and the y-axis represents the time taken for saving in milliseconds. The five different-colored lines represent the number of points collected in each individual acquisition. It can be observed that as the total number of data points required increases, the time taken for data saving by the acquisition card also increases accordingly. At the same time, the larger the number of points collected in each acquisition, the faster the corresponding acquisition is completed. Specifically, when the total number of data points is 10,000, the time taken for acquiring 500 points in a single acquisition is more than five times faster than acquiring 100 points in a single acquisition.

### 4.3. Validation of Ultrasonic Acquisition of the Chinese Fir

(1)Detection object: Chinese fir samples with an axial height of 6 cm and 7 cm, as shown in Figure 17 and Figure 18.(2)Detection direction: axial(3)Detection system parameters: as above

As can be seen from Table 2, the time of the first echo is about 0.0000121 s for axial ultrasound acquisition on a 6 cm high fir, and the velocity of ultrasound $v_{fir}$ is about 4958.678 m/s. Table 3 shows the axial ultrasonic measurement results of 7 cm Chinese fir. It can be concluded by calculation, that the echo time of the first ultrasonic signal is 0.0000138 s. It can be concluded by calculation, that the axial ultrasonic propagation speed is about 5090 m/s. The time-domain diagram of the measurement is shown in Figure 19.

A review of the literature [15], shows that the ultrasound propagation velocity in the cedar axial direction is between 4648.75–5593.83 m/s, and the experimentally calculated ultrasound velocity is within the plausible range.

### 4.4. Validation of Ultrasonic Acquisition of the Radermachera sinica

(1)Detection object: Chinese fir samples as shown in Figure 20(2)Detection direction: radial as shown in Figure 21(3)Detection system parameters: as above

Table 4 shows the measurement results. By conducting ten measurements, it was discovered that the average echo time of the first data point collected by the data acquisition card was 9.741 × 10^−6^ s, whereas the oscilloscope recorded an average echo time of 10.012 × 10^−6^ s for the first point. The average error of these ten experiments was approximately 0.271 microseconds. Furthermore, it is evident that the echo of the *Radermachera sinica* contains a substantial amount of information. However, collecting only 500 points as shown in Figure 22, does not allow for obtaining the complete echo information. In Figure 23, the complete collection of 4000 time domain diagrams, spectra and phase spectra are listed.

Through the above series of experiments, it is shown that the ultrasonic RF echo acquisition system designed in this paper can meet the function of ultrasonic acquisition of plant stems.

## 5. Discussion and Conclusions

Real-time nondestructive detection of plant stems is one of the hot topics in the study of plant physiological activities. Ultrasonic RF acquisition of plant stems allows for nondestructive and efficient plant physiological testing. Existing studies on the application of ultrasound to plants fall into three main categories: (1) the use of ultrasound measurements to calculate the physical and mechanical properties of plants [16,17], such as the modulus of elasticity of wood [18,19,20] and Young’s modulus [21]; (2) Studies on defect detection based on ultrasound technology, which can detect defects deep inside the plant [22,23]; (3) Plant anatomical studies based on ultrasound techniques [24,25] to understand the ultrasound representation of the anatomical properties of plants and to discover the relationship between the propagation of ultrasound in plants and their structure. The abovementioned application studies are studied by ultrasonic echo characteristics, so the ultrasound acquisition system is filtered on its RF signal in the high-frequency signal, only retaining the low-frequency signal to determine the time of the first echo position, while ignoring the time-varying characteristics of RF echo propagation in non-uniform plant stems; however, the present cultured plant experimental results show that the shape of the RF echo signal carries a lot of structural information about the stem, which is of great significance to the study of stem structure.

However, the proposed ultrasound acquisition system in this study also has certain limitations that should be addressed. One notable limitation is related to the data saving time, which can result in the omission of a small portion of data during the collection process. Ultrasonic waves experience attenuation and scattering during propagation, which limits their penetration depth. The limited penetration depth of ultrasound poses challenges in acquiring data from inhomogeneous media. In the case of plants with deep tissues, the penetration depth may not provide sufficient information. For instance, in this study, it was not possible to measure the radial supersonic velocity in excessively thick wood. Additionally, due to the parameter settings of the pulse generator, the radio frequency signal contains significant noise. Therefore, future research should focus on developing denoising techniques for the collected signal. Moreover, ultrasound acquisition generates a substantial amount of data, necessitating efficient data processing and algorithm implementation for real-time applications and large-scale data analysis. Achieving real-time and efficient processing of ultrasound signals presents a significant challenge. Future studies could explore the integration of ultrasound with other imaging modalities such as optical imaging and magnetic resonance imaging to obtain more comprehensive and detailed information. The advancement of multimodal imaging holds the potential to enhance image resolution and improve our understanding of tissue structures.

This paper completes the design of an ultrasound acquisition system for plant stems based on an ultrasound generator and receiver, high-speed data acquisition card, and LabVIEW custom control, realizing on-demand sample number setting, data acquisition, processing, display, and storage. This ensures that the location of the first echo acquired can be determined by the first layer of edge triggering when an ultrasound echo appears. For the detection of plant stems with excessive thickness, the use of the second layer of soft trigger acquisition to achieve the acquisition card, cannot effectively collect the signal of the complete echo due to capacity limitations. Through the velocity measurement and verification of ultrasound on organic glass and wood, the designed ultrasound acquisition system for plant stems is characterized by a short development cycle, high portability, low cost, simple data processing method, stable operation of the whole acquisition system, easy operation and simple interface, which can realize the demand of dynamic detection of plant stems.

## Figures and Tables

**Figure 1 sensors-23-07088-f001:**
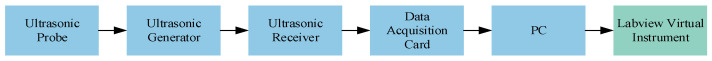
Ultrasonic RF acquisition system.

**Figure 3 sensors-23-07088-f003:**
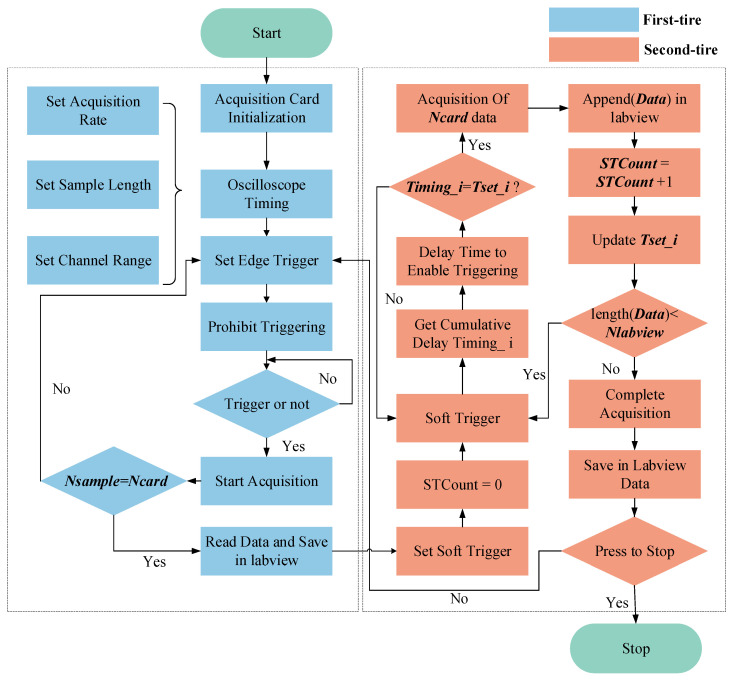
Sampling with double-layer multiple triggering.

**Figure 4 sensors-23-07088-f004:**
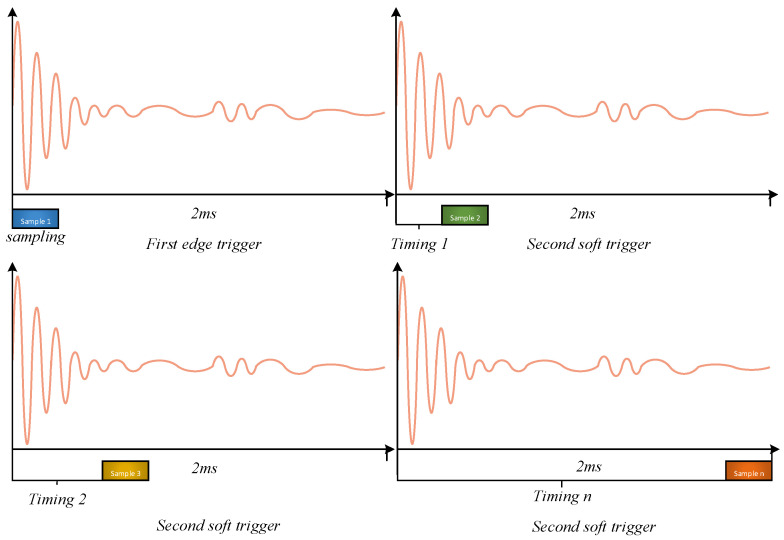
Double-layer multiple trigger sampling.

**Figure 5 sensors-23-07088-f005:**
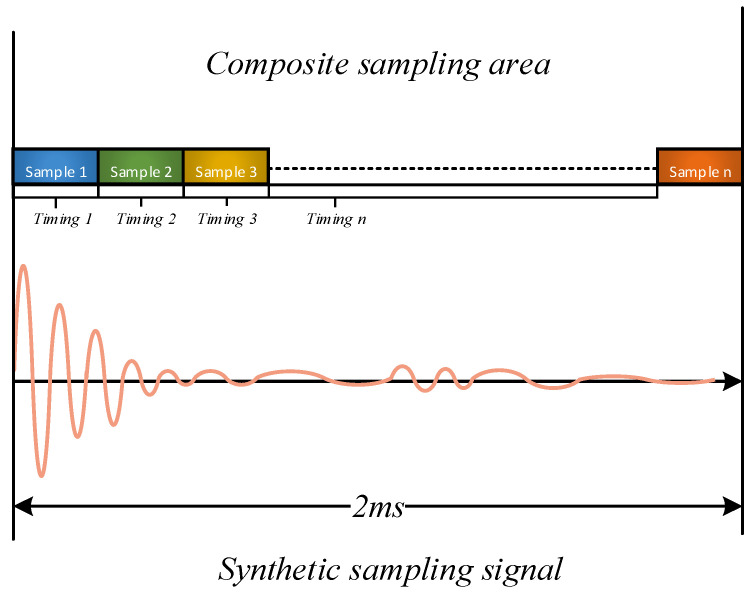
Synthetic sampling area and sampling signal.

**Figure 6 sensors-23-07088-f006:**
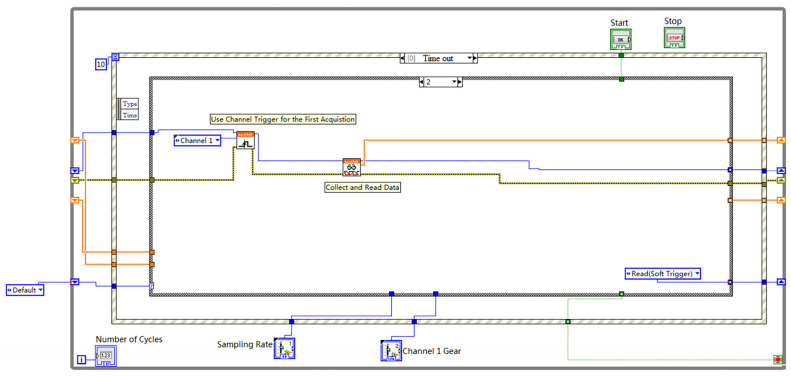
First layer rising edge triggered acquisition.

**Figure 7 sensors-23-07088-f007:**
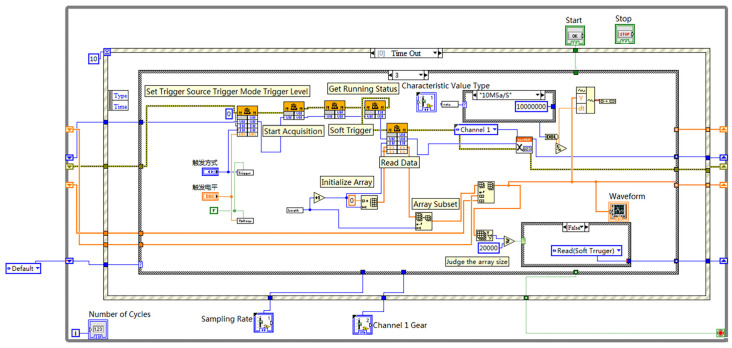
Second layer multiple timed trigger acquisition.

**Figure 8 sensors-23-07088-f008:**
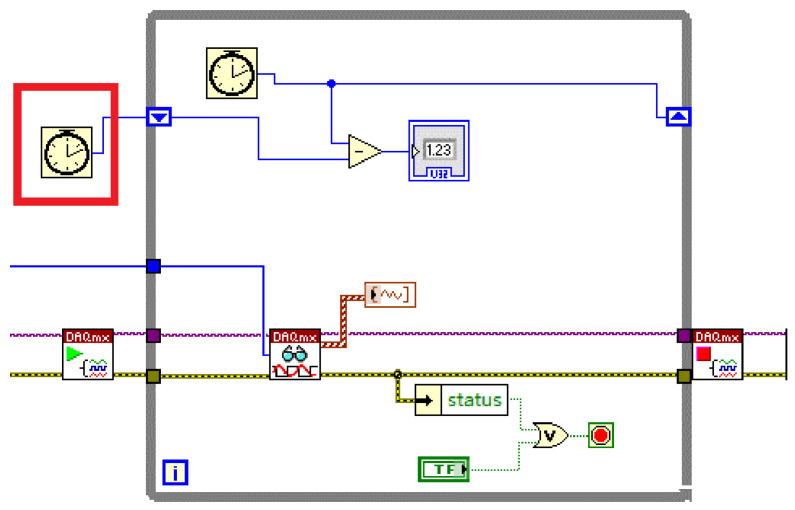
Timing soft trigger subroutine (The red box represents the Tint).

**Figure 9 sensors-23-07088-f009:**
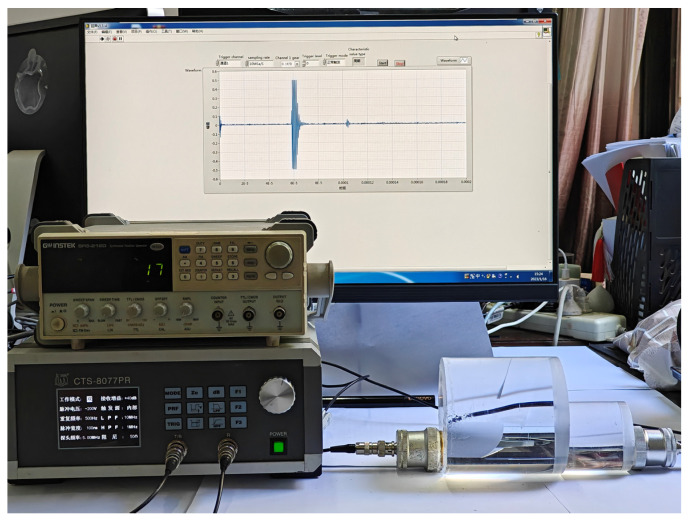
CTS-8077PR ultrasound generator-receiver and nextkit high-speed acquisition card.

**Figure 10 sensors-23-07088-f010:**
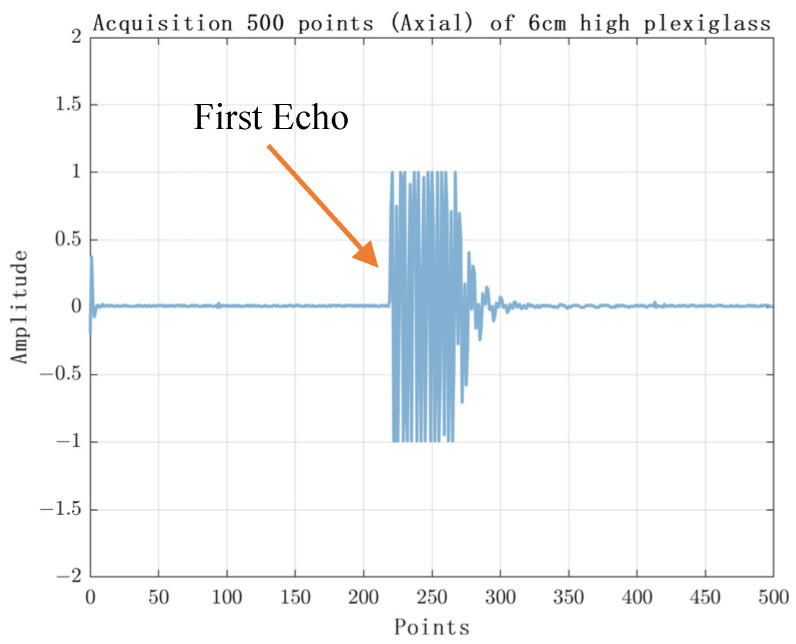
Single layer rising edge triggered acquisition of 500 samples.

**Figure 11 sensors-23-07088-f011:**
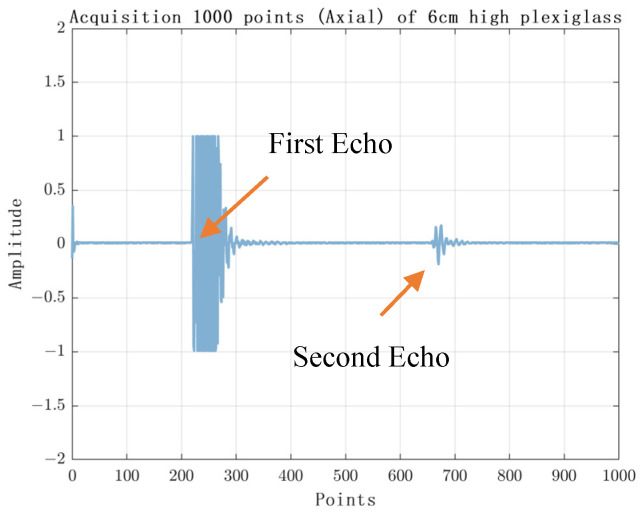
Double layer multiple trigger acquisition of 1000 samples.

**Figure 12 sensors-23-07088-f012:**
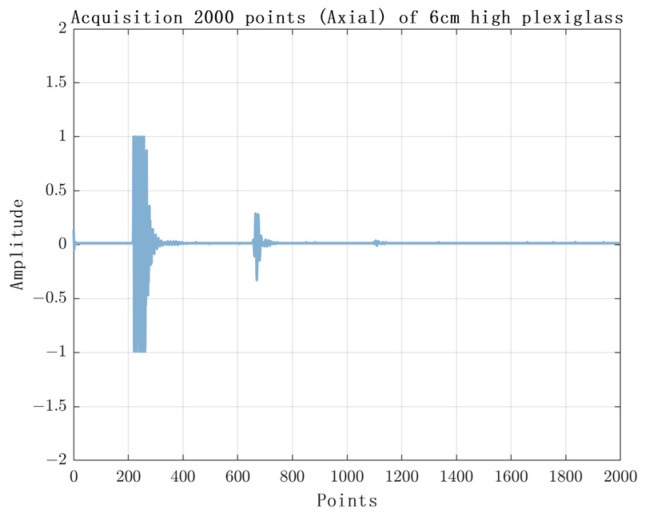
Single layer rising edge triggered acquisition of 2000 samples.

**Figure 13 sensors-23-07088-f013:**
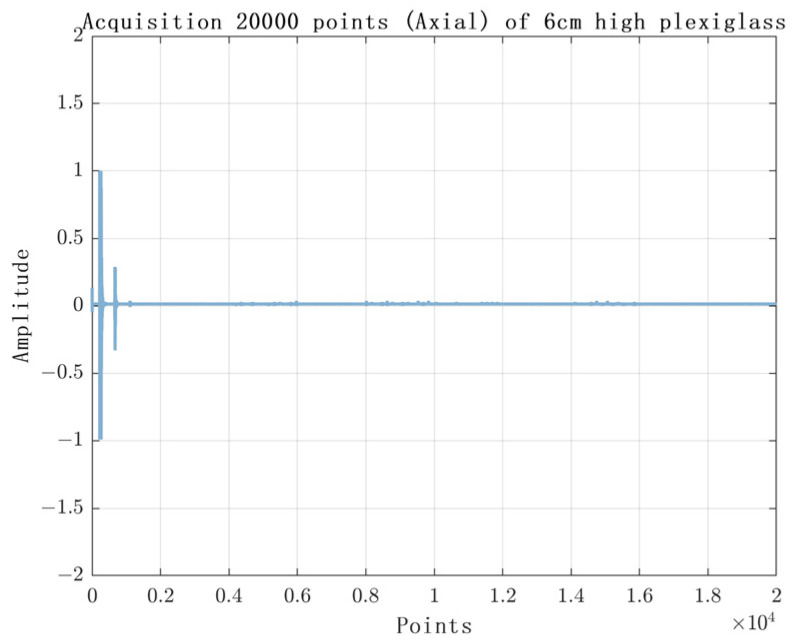
Single layer rising edge triggered acquisition of 20,000 samples.

**Figure 14 sensors-23-07088-f014:**
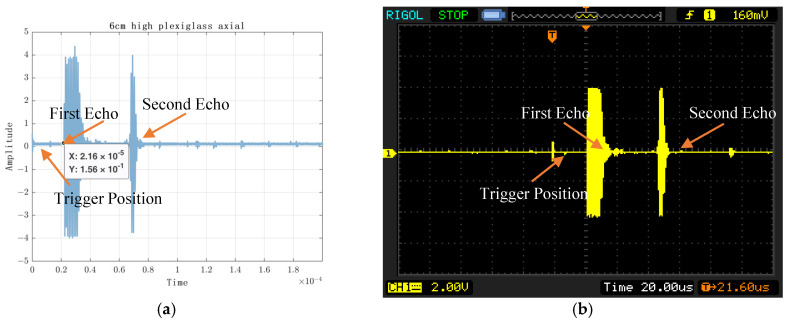
Plexiglass Ultrasonic RF Echo Signal Acquisition. (**a**) axial, 6 cm; (**b**) axial, 6 cm (oscillograph); (**c**) axial, 10 cm; (**d**) axial, 10 cm (oscillograph); (**e**) axial, 6 cm and 10 cm; (**f**) axial, 6 cm and 10 cm (oscillograph); (**g**) radial, 6 cm; (**h**) radial, 6 cm (oscillograph); (**i**) radial, 10 cm; (**j**) radial, 10 cm (oscillograph).

**Figure 15 sensors-23-07088-f015:**
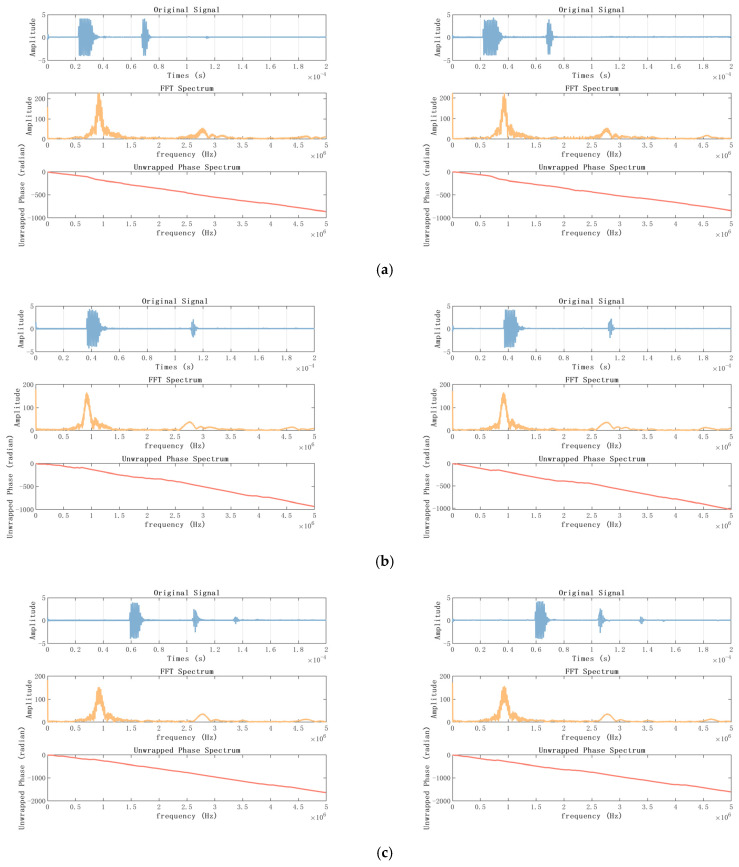
Time-domain diagram, spectrum and phase spectrum (on the left is the capture card, on the right is the oscilloscope). (**a**) axial, 6 cm; (**b**) axial, 10 cm; (**c**) axial, 6 cm and 10 cm; (**d**) radial, 6 cm; (**e**) radial, 10 cm.

**Figure 16 sensors-23-07088-f016:**
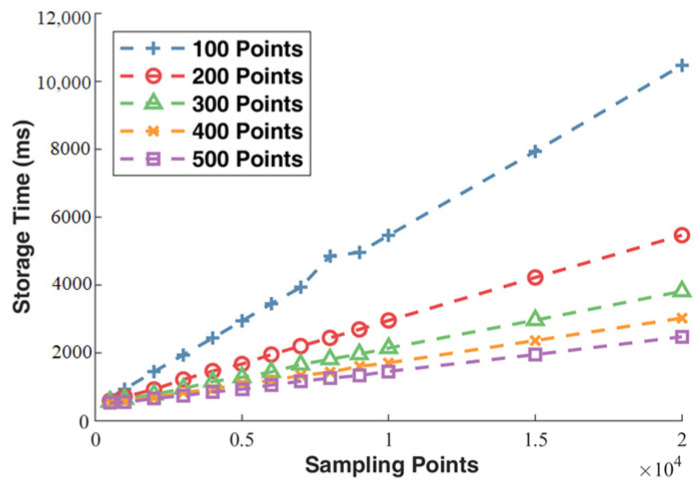
Number of Data Points vs. Recording Speed.

**Figure 17 sensors-23-07088-f017:**
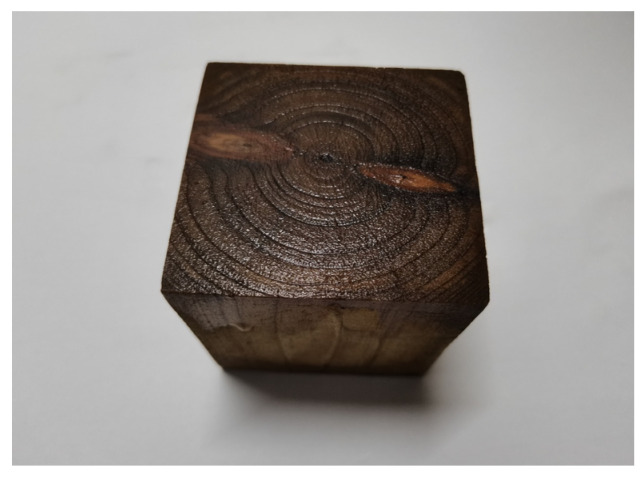
Fir wood test samples of 6 cm.

**Figure 18 sensors-23-07088-f018:**
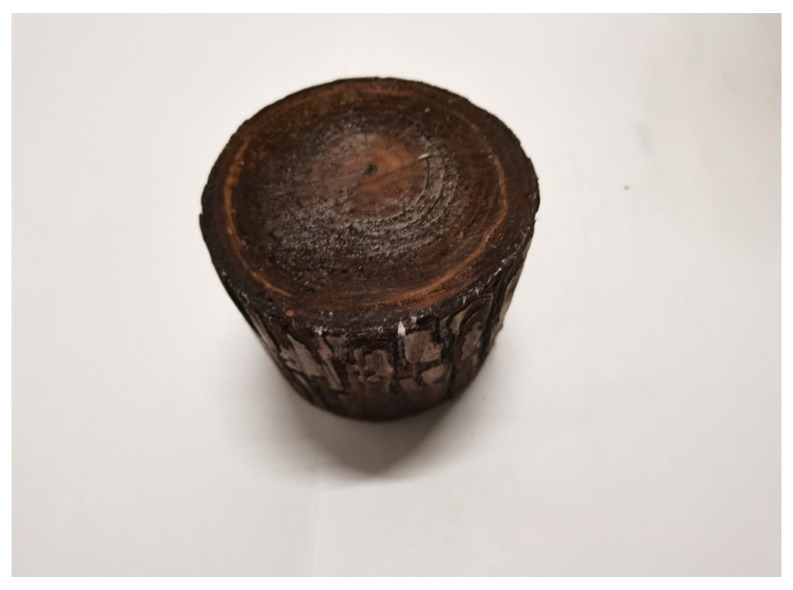
Fir wood test samples of 7 cm.

**Figure 19 sensors-23-07088-f019:**
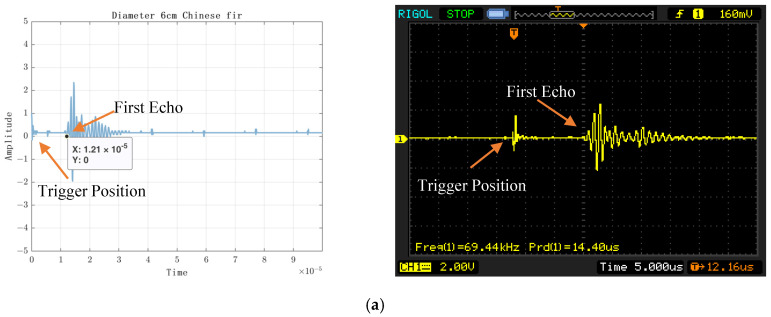
Chinese fir time-domain waveform (with acquisition card on the left and oscilloscope on the right). (**a**) 6 cm, (**b**) 7 cm.

**Figure 20 sensors-23-07088-f020:**
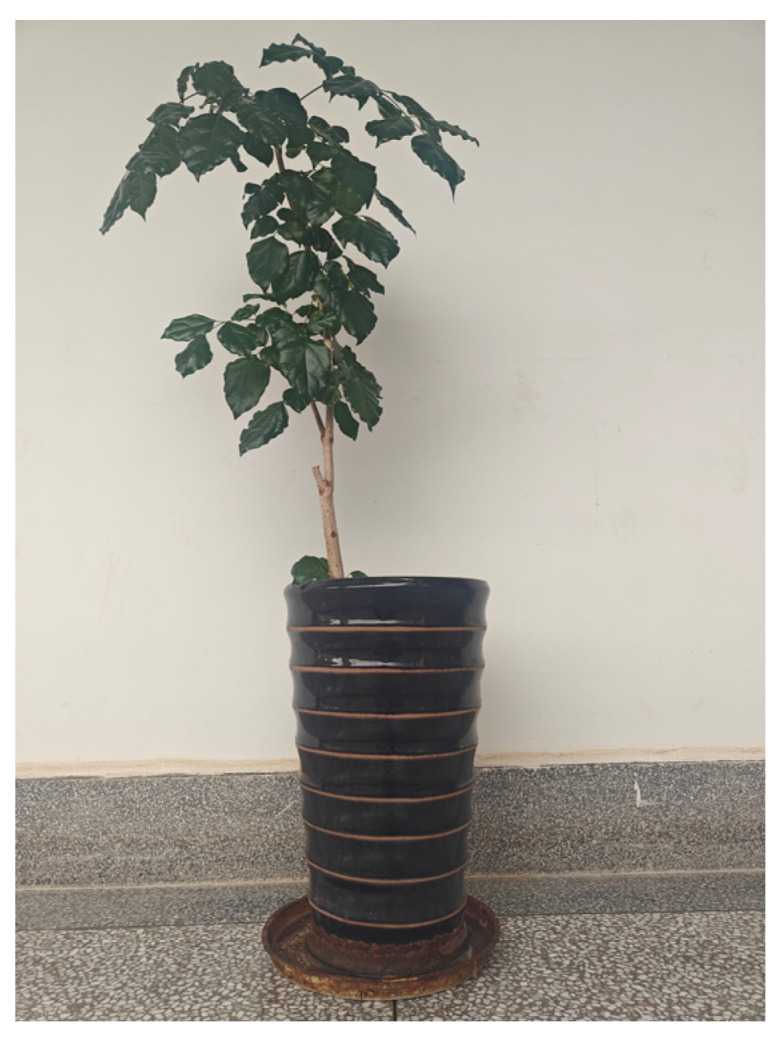
The *Radermachera sinica* sample.

**Figure 21 sensors-23-07088-f021:**
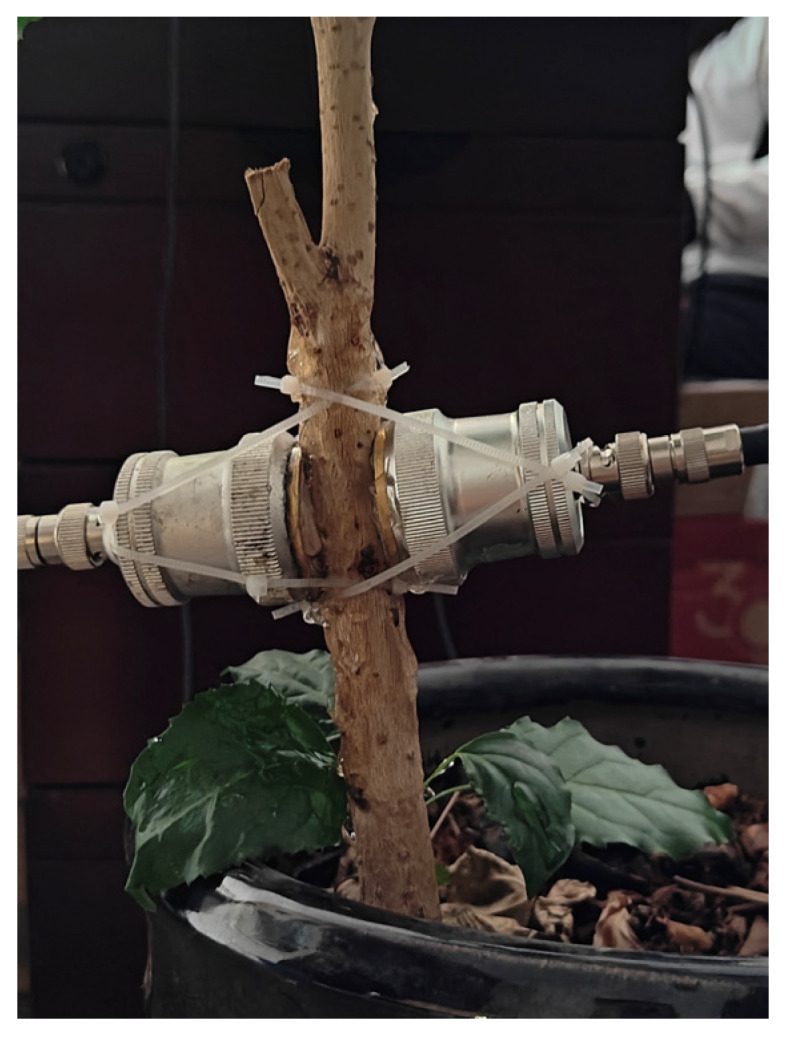
Measuring plant stems.

**Figure 22 sensors-23-07088-f022:**
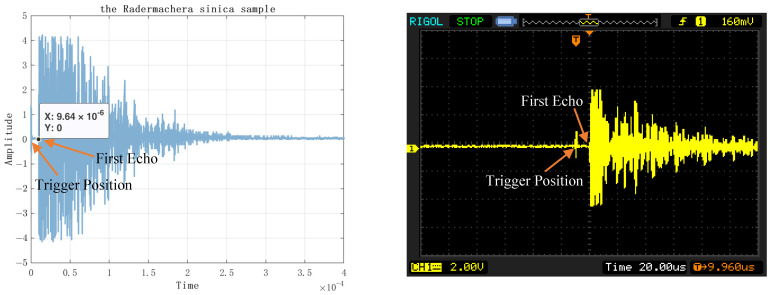
The *Radermachera sinica* time-domain waveform (with acquisition card on the left and oscilloscope on the right).

**Figure 23 sensors-23-07088-f023:**
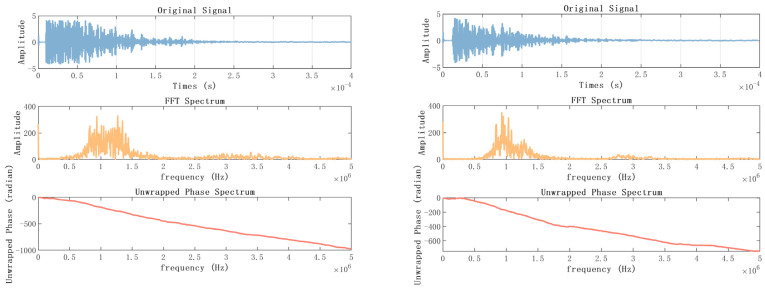
Time-domain diagram, spectrum and phase spectrum of the *Radermachera sinica* (with acquisition card on the left and oscilloscope on the right).

**Table 1 sensors-23-07088-t001:** Measurement of echo time and calculated ultrasound propagation velocity of plexiglass.

Actual Thickness	Collection Direction	First Echo Signal Time (DAQ) (s)	First Echo Signal Time (Oscillograph) (s)	Detection of Thickness (cm)
6 cm	Axial	2.14 × 10^−5^ ± 1.56 × 10^−7^	2.21 × 10^−5^ ± 1.67 × 10^−7^	6.08 × 10^0^ ± 1.99 × 10^−2^
	Radial	2.12 × 10^−5^ ± 1.33 × 10^−7^	2.15 × 10^−5^ ± 1.47 × 10^−7^	6.07 × 10^0^ ± 1.52 × 10^−2^
10 cm	Axial	3.64 × 10^−5^ ± 9.17 × 10^−8^	3.65 × 10^−5^ ± 4.77 × 10^−7^	1.01 × 10^1^ ± 1.25 × 10^−2^
	Radial	3.66 × 10^−5^ ± 9.80 × 10^−8^	3.65 × 10^−5^ ± 4.80 × 10^−7^	1.01 × 10^1^ ± 1.34 × 10^−2^
6 cm + 10 cm	Axial	5.87 × 10^−5^ ± 9.80 × 10^−8^	5.88 × 10^−5^ ± 1.84 × 10^−7^	1.61 × 10^1^ ± 1.34 × 10^−2^

**Table 2 sensors-23-07088-t002:** Axial echo time and calculated ultrasound propagation velocity of 6 cm cedar wood.

Number of Times	Axial Ultrasound First Echo Signal Time (s)	Axial Speed (0 m/s)
1	1.21 × 10^−5^	4.96 × 10^3^
2	1.20 × 10^−5^	5.00 × 10^3^
3	1.21 × 10^−5^	4.96 × 10^3^
4	1.21 × 10^−5^	4.96 × 10^3^
5	1.21 × 10^−5^	4.91 × 10^3^
6	1.20 × 10^−5^	5.00 × 10^3^
7	1.20 × 10^−5^	5.00 × 10^3^
8	1.22 × 10^−5^	4.91 × 10^3^
9	1.21 × 10^−5^	4.96 × 10^3^
10	1.21 × 10^−5^	4.96 × 10^3^

**Table 3 sensors-23-07088-t003:** Axial echo time and calculated ultrasound propagation velocity of 7 cm cedar wood.

Number of Times	Axial Ultrasound First Echo Signal Time (s)	Axial Speed (m/s)
1	1.39 × 10^−5^	5.04 × 10^3^
2	1.39 × 10^−5^	5.04 × 10^3^
3	1.36 × 10^−5^	5.15 × 10^3^
4	1.35 × 10^−5^	5.19 × 10^3^
5	1.41 × 10^−5^	4.96 × 10^3^
6	1.41 × 10^−5^	4.96 × 10^3^
7	1.40 × 10^−5^	5.00 × 10^3^
8	1.34 × 10^−5^	5.22 × 10^3^
9	1.35 × 10^−5^	5.19 × 10^3^
10	1.36 × 10^−5^	5.15 × 10^3^

**Table 4 sensors-23-07088-t004:** Axial echo time and calculated ultrasound propagation velocity of the *Radermachera sinica*.

Number of Times	First Echo Signal Time (s)	First Echo Signal Time (Oscillograph) (s)
1	9.92 × 10^−6^	9.96 × 10^−6^
2	9.76 × 10^−6^	9.76 × 10^−6^
3	9.60 × 10^−6^	10.24 × 10^−6^
4	9.41 × 10^−6^	10.04 × 10^−6^
5	9.90 × 10^−6^	10.32 × 10^−6^
6	9.90 × 10^−6^	9.76 × 10^−6^
7	9.76 × 10^−6^	10.17 × 10^−6^
8	9.54 × 10^−6^	9.68 × 10^−6^
9	9.86 × 10^−6^	10.52 × 10^−6^
10	9.76 × 10^−6^	9.67 × 10^−6^

## Data Availability

Not applicable.

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
