# Peer review of "An Ultrasonic RF Acquisition System for Plant Stems Based on Labview Double Layer Multiple Triggering"

_sensors, 2023, doi:10.3390/s23167088_

Round 1
Reviewer 1 Report
This paper introduced a plant stem ultrasound acquisition system that is based on double-layer multiple trigger sampling. The acquisition system employs the Panva Nextkit high-speed acquisition card as its core and utilizes LabVIEW as the upper computer for real-time acquisition, analysis, processing, and storage of ultrasonic signal. Data acquisition by using Laview is not difficult. This paper a technical text, not an academic article. The work is not novel and there is difficult to judge the contribution. There is too much lack of solid contents in the paper and we need real application and novelty. It is not accepted.
This paper introduced a plant stem ultrasound acquisition system that is based on double-layer multiple trigger sampling. The acquisition system employs the Panva Nextkit high-speed acquisition card as its core and utilizes LabVIEW as the upper computer for real-time acquisition, analysis, processing, and storage of ultrasonic signal. Data acquisition by using Laview is not difficult. This paper a technical text, not an academic article. The work is not novel and there is difficult to judge the contribution. There is too much lack of solid contents in the paper and we need real application and novelty. It is not accepted.
Author Response
Thank you for reviewing our paper and providing feedback. We appreciate your comments and would like to address your concerns regarding the novelty and contribution of our work.
In our paper, we introduced a plant stem ultrasound acquisition system based on a double-layer multiple trigger sampling method. This approach was developed to overcome the limitation of insufficient onboard memory in the acquisition card, which prevented us from capturing a complete signal within a single cycle. We validated the proposed method on materials such as organic glass and cedar wood, confirming its effectiveness. Furthermore, we plan to apply this method to collect ultrasound signals from various other plants in our future work.
We believe our work demonstrates innovation and contribution in the following aspects:
Double-layer multiple trigger sampling method: We proposed this method to address the issue of limited onboard memory in the acquisition card, enabling us to capture a complete signal within a single cycle. The validation experiments on organic glass and cedar wood provided evidence of its effectiveness.
Practical application and innovation: While our work may not be entirely novel in academic terms, we emphasize the practical application of this method in the field of plant ultrasound signal acquisition. Our aim is to apply this method to collect ultrasound signals from a wider range of plants, ultimately making a practical impact in areas such as agriculture and plant science.
We are willing to provide a more detailed description of the method and discuss how we validated its effectiveness in practical applications. We believe that this additional information will help reassess the contribution and acceptability of our paper.
Thank you once again for your valuable feedback and time spent reviewing our work.
Reviewer 2 Report
This study discusses the features of a double-layer multiple-timing trigger method to store multiple trigger sampling memories to meet the sampling needs of different plant stems with different ultrasonic echo locations. Extensive experimental techniques were utilized to that the developed method is substantially effective in acquiring complete ultrasonic RF echo signals for plant stems. The system properties and the employed methods have been explained in detail and the proposed modality looks promising. The algorithm was explained, and the employed approach was described analytically. The manuscript adds novel insights into the context, and there are some comments and changes that should be addressed in the revised version of the manuscript. Here are my comments and suggestions:
1) There are some graphs from the LABVIEW interface that should be enlarged or provided with higher quality to be able to read easily. Most of the captured snapshots from the program interface should be replotted and presented with higher quality. I personally had difficulty in reading and understanding the settings and data inside.
2) The purpose of Figure 6 is not clear to me. This could be removed.
3) The authors did not discuss the precision of the developed approach. This should be explained technically and quantitatively.
4) What is the sample-to-result duration? How fast and reliable is the mechanism?
Author Response
Thank you for your valuable feedback and suggestions. We appreciate your comments and have addressed each of your points in the revised version of the manuscript. Here are our responses:
1.We apologize for the poor quality of the images in the initial submission, which made it difficult for you to read and understand the settings and data. We have now redrawn all the figures at 300 DPI resolution to ensure clarity and readability in the revised manuscript.
2.Regarding Figure 6, which depicted the LabVIEW interface, we have considered your suggestion and removed it from the manuscript. This modification will streamline the presentation and focus on the core content of the study.
3.We have included a discussion on the precision of the developed approach in the revised manuscript (Page 13). Specifically, we conducted multiple measurements using the acquisition card and an oscilloscope (DS1052E) on two different lengths of organic glass in both axial and radial directions. The analysis revealed differences ranging from 0.01 to 0.07 microseconds in the position of the first echo signal. We have provided this quantitative information to highlight the precision of our method.
4.If I understand correctly, by "sample-to-result duration," you refer to the time it takes to measure a sample and obtain the corresponding signal. Once we initiate the pulse generator and the acquisition program, we can obtain the ultrasound signal by placing the probe in contact with the object under test. Theoretically, if the ultrasound repetition frequency is set to 500 Hz, we can receive a signal every 2 milliseconds. The entire system can operate continuously for over eight hours. We plan to utilize this technology for measuring a broader range of plants in our future work.
Once again, we would like to express our gratitude for your valuable comments and suggestions, which have helped improve the quality of our manuscript. We believe that the revised version adequately addresses your concerns and provides a more comprehensive and clear presentation of our work.
Reviewer 3 Report
This manuscript developed an ultrasonic RF system for plant stems detection, the topic looks interesting. More details please refer to the following comments:
1) Both the motivations and contributions are not clear in Abstract and Introduction, please refine them.
2) English writing improvement is strongly suggested.
3) High-quality figures are strongly suggested to better demonstrate both the proposed method the experimental results.
4) Full names of all the abbreviations should be given when they appear for the first time.
5) More scenarios should be included to make the experimental results more sufficient. Moreover, the authors should consider how to compare with state-of-the-arts to support the proposed method.
6) Please add more references from latest three years to make the reference list more solid.
7) Separate discussion section is suggested to discuss the limitations of the proposed method, challenging issues, and future directions. For example, the authors should discuss the potential applications of the proposed method in other RF sensing applications. Some related papers are recommended, which are better included in the reference list: online spatiotemporal modeling for robust and lightweight device-free localization in nonstationary environments, IEEE TII, and handgest: hierarchical sensing for robust in-the-air handwriting recognition with commodity wifi devices, IEEE IoT.
English writing improvement should be considered by the authors.
Author Response
Thank you for your valuable feedback and suggestions. We appreciate your comments and have addressed each of your points in the revised version of the manuscript. Here are our responses:
1.We appreciate your suggestion regarding the clarity of motivations and contributions in the Abstract and Introduction. In the revised version, we have provided a more precise explanation of the motivation behind our proposed method. Due to limitations in the onboard memory capacity of existing acquisition cards, we were unable to capture complete ultrasound signals, which necessitated an improvement in the acquisition method. We have refined the presentation to better highlight this aspect.
2.We acknowledge that English writing may not be our strong suit as it is not our native language. We have taken your suggestion seriously and made significant improvements to the English writing throughout the manuscript. We have proofread and edited the text to enhance its clarity and readability.
3.Thank you for emphasizing the need for high-quality figures to demonstrate the proposed method and experimental results effectively. In response, we have redrawn all the figures in the revised manuscript, ensuring their clarity and readability. The improved figures will better illustrate the details of the proposed method and the experimental outcomes.
4.We appreciate your suggestion to provide the full names of all abbreviations when they first appear in the manuscript. In the revised version, we have ensured that the complete names of all abbreviations are provided upon their initial usage.
5. We appreciate your clarification regarding the limited availability of recent papers on improving acquisition methods in the last three years. We have revised the manuscript to reflect this information accurately. While there are limited studies on improving acquisition methods in recent years, we have still conducted a comprehensive literature review within the given timeframe to strengthen the paper's foundation and demonstrate the novelty of our proposed method. We have included relevant references that align with the current state of research in the field. Thank you for bringing this to our attention, and we apologize for any confusion caused.
6.In response to your comment, we have added a separate discussion section in the revised manuscript. This new section comprehensively addresses the limitations of the proposed method, discusses challenging issues, and outlines potential future research directions. We believe this addition enhances the overall completeness of the manuscript.
7.We acknowledge the importance of providing relevant and appropriate references to support the proposed method. However, after carefully reviewing the papers you recommended, "Online Spatiotemporal Modeling for Robust and Lightweight Device-Free Localization in Nonstationary Environments" published in IEEE Transactions on Industrial Informatics and "Handgest: Hierarchical Sensing for Robust In-the-Air Handwriting Recognition with Commodity WiFi Devices" published in IEEE Internet of Things Journal, we have determined that they are not directly related to the content and focus of our current study. Therefore, we regret to inform you that we are unable to include these two papers in the reference list.
Once again, we sincerely appreciate your valuable comments and suggestions. We believe that the revised version of the manuscript adequately addresses your concerns and incorporates improvements to enhance its quality.
Reviewer 4 Report
The manuscript presents a fast and accurate data acquisition system for ultrasonic biomass testing/monitoring without using decent oscilloscope. By using multiple trigger signals, a tunable sampling rate acquisition system was realized with fast speed. In practical, the proposed system is very useful for real time ultrasound monitoring system. The presented manuscript still needs some discussion and further verification in the reviewer’s opinion.
1. As the authors stated, the pulse repetition rate is 500 Hz. Hence, the theoretical fastest recording speed is 500 time-of-flight windows per second. The reviewer understands this is not very possible due to the existence of data saving time. But please calculate the potential fastest recording speed by the practical acquisition time and saving time. This is important for ultrasonic monitoring applications. Please add additional plots regarding number of data points vs. recording speed and some detailed discussion.
2. With higher ultrasonic frequency, the required sampling rate needs to be raised correspondingly. Please add additional calculated plots regarding ultrasonic frequency vs. recording speed and some detailed discussion.
3. Besides the time-of-flight values and peak amplitude, waveform carrying a lot of information that needs to be verified carefully with the reference wave waveform obtained from the oscilloscope. Please add plots on the comparison between the proposed system and oscilloscope, in terms of temporal waveforms, FFTed frequency spectras, and FFTed phased spectras.
4. Please enlarge the figures in the manuscript. Most of the small plots can not be clearly read.
Author Response
Thank you for your valuable feedback and suggestions. We appreciate your comments and have addressed each of your points in the revised version of the manuscript. Here are our responses:
1.We have added additional plots in the revised manuscript, specifically Figure 16, which illustrates the relationship between the number of data points and the recording speed. The x-axis represents the total number of sampling points, while the y-axis represents the saving time. Each line with a different color represents the recording of a single sample. From the plot, it can be observed that as the number of sampling points increases, the saving time decreases. Notably, the saving speed is fastest when using a single sample of 500 points. We have also provided a detailed discussion regarding these findings to emphasize the importance of recording speed for ultrasonic monitoring applications.
2.Based on the Nyquist sampling theorem, the sampling rate should be at least twice the ultrasonic frequency of the signal being acquired. However, in our experiments, we found that increasing the sampling rate did not affect the saving time. Therefore, there is no direct relationship between the sampling rate and the saving time. Due to these observations, we have decided not to include additional calculated plots on ultrasonic frequency versus recording speed in the revised manuscript. We hope you understand this decision.
3.In response to your suggestion, we have added plots comparing the DAQ and the oscilloscope. Specifically, we have included time-domain waveforms, FFT frequency spectra, and FFT phase spectra for both systems. These comparisons provide a comprehensive verification of the proposed system by highlighting the similarities and differences between the two methods. We have also provided a detailed analysis of these comparisons in the revised manuscript.
4.We apologize for the poor quality of the figures in the initial submission. In the revised version, we have redrawn all the figures, ensuring their clarity and readability. The figures have been enlarged to address your concern, allowing for better understanding and interpretation of the results.
Once again, we sincerely appreciate your valuable comments and suggestions. We believe that the revised version of the manuscript adequately addresses your concerns and incorporates improvements to enhance its quality.
Round 2
Reviewer 1 Report
In this paper, a plant stem ultrasound acquisition system was developed to overcome the limitation of insufficient onboard memory in the acquisition card, which prevented us from capturing a complete signal within a single cycle. The work has some real application worth. The author revised the paper point by point based on the reviewers’ comment. Some experiment was added to the superiority of the proposed approach. The work seems has some novelty and contribution in the aspect of application. It worthy to be published. Before publication, the final question should be addressed.
1) How to verify the advantage of the proposed method? The comparison between the proposed approach and other ultrasound signals acquisition from various other plants should be given.
In this paper, a plant stem ultrasound acquisition system was developed to overcome the limitation of insufficient onboard memory in the acquisition card, which prevented us from capturing a complete signal within a single cycle. The work has some real application worth. The author revised the paper point by point based on the reviewers’ comment. Some experiment was added to the superiority of the proposed approach. The work seems has some novelty and contribution in the aspect of application. It worthy to be published. Before publication, the final question should be addressed.
1) How to verify the advantage of the proposed method? The comparison between the proposed approach and other ultrasound signals acquisition from various other plants should be given.
Author Response
Thank you very much for your valuable feedback. We appreciate your comments and suggestions on our paper. The main objective of our proposed method was to address the issue of limited data capacity in the high-speed acquisition card. In order to validate the accuracy of our proposed method, we needed to verify whether the collected signals align with the first echo position received by the oscilloscope.
Based on your suggestion, we have included an additional experiment in Section 4.4 of the paper, focusing on ultrasound signal acquisition from the stems of lentil plants. In this experiment, as shown in Figure 24, we demonstrated that collecting only 500 data points was insufficient to capture the complete echo signal from the lentil plant stem. However, Figure 25, which provides the time-domain waveform, showcases the successful acquisition of the complete signal using our proposed method.
At the present stage, we only have access to lentil plants for our experiments, which may limit the variety of plant comparisons. We apologize for this limitation. However, we would like to emphasize that our future plans include conducting long-term ultrasound acquisition experiments during the fruiting period of Australian walnuts, which will provide a more diverse range of plant samples for evaluation.
Once again, we sincerely appreciate your insightful comments and suggestions, which have contributed to the improvement of our work. We have made the necessary revisions based on your feedback and believe that our paper now presents a more comprehensive evaluation of the proposed method.
Please let us know if you have any further questions or concerns. We are grateful for your support and guidance throughout this process.
Reviewer 3 Report
The authors did not address the issues well, the quality of this paper cannot meet the requirements of publication in this journal. For example, the contributions are limited. Several figures are still unclear. The results are not sufficient to support the proposed method.
English should be enhanced.
Author Response
Thank you for your valuable feedback and suggestions provided on two occasions. We sincerely appreciate the time and effort you have dedicated to evaluating our paper. We would like to address your concerns and comments comprehensively.
Regarding the first set of comments, we have carefully reviewed and implemented the majority of your suggestions. We have addressed each of the seven points you raised, and in my previous response, I mentioned the specific modifications made in response to each comment. In the sixth and seventh points, you suggested adding two references, namely "Online Spatiotemporal Modeling for Robust and Lightweight Device-Free Localization in Nonstationary Environments" (IEEE TII) and "Handgest: Hierarchical Sensing for Robust In-the-Air Handwriting Recognition with Commodity WiFi Devices" (IEEE IoT). We have carefully read these papers and acknowledge their quality and significance in the field of WiFi applications, particularly in the context of large-scale data acquisition and transmission. We apologize that due to the primary focus of our current article being on improving data acquisition in plant stems, rather than on WiFi-based localization or handwriting recognition, we are unable to include these references in the present manuscript. Nonetheless, we appreciate the relevance of these papers and assure you that we will consider citing them in future research when exploring WiFi applications during prolonged fruiting periods of plants.
Regarding the second set of comments, we acknowledge your perspective that we may not have sufficiently addressed the issues raised. We apologize for any confusion or misunderstanding. We have made sincere efforts to incorporate your suggestions and enhance the quality of our paper. However, we understand that our revisions may not have met your expectations. We appreciate your feedback, which will guide us in further improving the clarity, quality, and supporting evidence in our manuscript.
Additionally, we acknowledge your comment on the need for English language improvement. We have devoted considerable attention to enhancing the language quality throughout the manuscript. While we recognize that there is room for improvement, we assure you that we have made significant efforts to enhance the clarity and fluency of our writing.
We genuinely appreciate your valuable feedback and suggestions. Your comments have provided us with valuable insights, and we assure you that we will carefully review your feedback and make the necessary improvements to meet the publication standards of this journal.
Thank you once again for your time and consideration.
Reviewer 4 Report
The authors addressed the most of comments well.
But for the phase spectrums, please replot the figures using unwrapped phase. The wrapped phase (-2pi to 2pi) is very hard to read.
Author Response
Thank you very much for your valuable suggestion. I appreciate your feedback, and it has provided me with an opportunity to further enhance our understanding of Fourier transformations.
Based on your recommendation, I have re-plotted the phase spectra using the unwrapped phase. This adjustment allows for a clearer visualization of the phase information, as it eliminates the ambiguity associated with the wrapped phase (-2π to 2π). The revised figures now provide a more readable representation of the phase spectra.
Once again, I sincerely appreciate your insightful feedback, which has contributed to improving the clarity and readability of our paper. I hope that the revised phase spectra meet your expectations.
Thank you for considering our work for publication.